# Linear Convergence with Condition Number Independent Access of Full Gradients

**Lijun Zhang     Mehrdad Mahdavi     Rong Jin**
Department of Computer Science and Engineering
Michigan State University, East Lansing, MI 48824, USA
{zhanglij,mahdavim,rongjin}@msu.edu

## Abstract

For smooth and strongly convex optimizations, the optimal iteration complexity of the gradient-based algorithm is $O(\sqrt{\kappa} \log 1/\epsilon)$, where $\kappa$ is the condition number. In the case that the optimization problem is ill-conditioned, we need to evaluate a large number of full gradients, which could be computationally expensive. In this paper, we propose to remove the dependence on the condition number by allowing the algorithm to access stochastic gradients of the objective function. To this end, we present a novel algorithm named Epoch Mixed Gradient Descent (EMGD) that is able to utilize two kinds of gradients. A distinctive step in EMGD is the mixed gradient descent, where we use a combination of the full and stochastic gradients to update the intermediate solution. Theoretical analysis shows that EMGD is able to find an $\epsilon$-optimal solution by computing $O(\log 1/\epsilon)$ full gradients and $O(\kappa^2 \log 1/\epsilon)$ stochastic gradients.

## 1   Introduction

Convex optimization has become a tool central to many areas of engineering and applied sciences, such as signal processing [20] and machine learning [24]. The problem of convex optimization is typically given as

$$\min_{\mathbf{w} \in \mathcal{W}} F(\mathbf{w}),$$

where $\mathcal{W}$ is a convex domain, and $F(\cdot)$ is a convex function. In most cases, the optimization algorithm for solving the above problem is an iterative process, and the convergence rate is characterized by the iteration complexity, i.e., the number of iterations needed to find an $\epsilon$-optimal solution [3, 17]. In this study, we focus on first order methods, where we only have the access to the (stochastic) gradient of the objective function. For most convex optimization problems, the iteration complexity of an optimization algorithm depends on the following two factors.

1. The analytical properties of the objective function. For example, is $F(\cdot)$ smooth or strongly convex?

2. The information that can be elicited about the objective function. For example, do we have access to the full gradient or the stochastic gradient of $F(\cdot)$?

The optimal iteration complexities for some popular combinations of the above two factors are summarized in Table 1 and elaborated in the related work section. We observe that when the objective function is smooth (and strongly convex), the convergence rate for full gradients is much faster than that for stochastic gradients. On the other hand, the evaluation of a stochastic gradient is usually significantly more efficient than that of a full gradient. Thus, replacing full gradients with stochastic gradients essentially trades the number of iterations with a low computational cost per iteration.

Table 1: The optimal iteration complexity of convex optimization. $L$ and $\lambda$ are the moduli of smoothness and strong convexity, respectively. $\kappa = L/\lambda$ is the condition number.

|  | Lipschitz continuous | Smooth | Smooth & Strongly Convex |
|---|---|---|---|
| Full Gradient | $O\left(\frac{1}{\epsilon^2}\right)$ | $O\left(\frac{L}{\sqrt{\epsilon}}\right)$ | $O\left(\sqrt{\kappa}\log\frac{1}{\epsilon}\right)$ |
| Stochastic Gradient | $O\left(\frac{1}{\epsilon^2}\right)$ | $O\left(\frac{1}{\epsilon^2}\right)$ | $O\left(\frac{1}{\lambda\epsilon}\right)$ |

In this work, we consider the case when the objective function is both smooth and strongly convex, where the *optimal* iteration complexity is $O(\sqrt{\kappa}\log\frac{1}{\epsilon})$ if the optimization method is first order and has access to the full gradients [17]. For the optimization problems that are ill-conditioned, the condition number $\kappa$ can be very large, leading to many evaluations of full gradients, an operation that is computationally expensive for large data sets. To reduce the computational cost, we are interested in the possibility of making the number of full gradients required independent from $\kappa$. Although the $O(\sqrt{\kappa}\log\frac{1}{\epsilon})$ rate is in general not improvable for any first order method, we bypass this difficulty by allowing the algorithm to have access to both full and stochastic gradients. Our objective is to reduce the iteration complexity from $O(\sqrt{\kappa}\log\frac{1}{\epsilon})$ to $O(\log\frac{1}{\epsilon})$ by replacing most of the evaluations of full gradients with the evaluations of stochastic gradients. Under the assumption that stochastic gradients can be computed efficiently, this tradeoff could lead to a significant improvement in computational efficiency.

To this end, we developed a novel optimization algorithm named Epoch Mixed Gradient Descent (EMGD). It divides the optimization process into a sequence of epochs, an idea that is borrowed from the epoch gradient descent [9]. At each epoch, the proposed algorithm performs *mixed gradient descent* by evaluating *one* full gradient and $O(\kappa^2)$ stochastic gradients. It achieves a constant reduction in the optimization error for every epoch, leading to a linear convergence rate. Our analysis shows that EMGD is able to find an $\epsilon$-optimal solution by computing $O(\log\frac{1}{\epsilon})$ full gradients and $O(\kappa^2\log\frac{1}{\epsilon})$ stochastic gradients. In other words, with the help of stochastic gradients, the number of full gradients required is reduced from $O(\sqrt{\kappa}\log\frac{1}{\epsilon})$ to $O(\log\frac{1}{\epsilon})$, independent from the condition number.

## 2 Related Work

During the last three decades, there have been significant advances in convex optimization [3,15,17]. In this section, we provide a brief review of the first order optimization methods.

We first discuss deterministic optimization, where the gradient of the objective function is available. For the general convex and Lipschitz continuous optimization problem, the iteration complexity of gradient (subgradient) descent is $O(\frac{1}{\epsilon^2})$, which is optimal up to constant factors [15]. When the objective function is convex and smooth, the optimal optimization scheme is the accelerated gradient descent developed by Nesterov, whose iteration complexity is $O(\frac{L}{\sqrt{\epsilon}})$ [16, 18]. With slight modifications, the accelerated gradient descent algorithm can also be applied to optimize the smooth and strongly convex objective function, whose iteration complexity is $O(\sqrt{\kappa}\log\frac{1}{\epsilon})$ and is in general not improvable [17, 19]. The objective of our work is to reduce the number of accesses to the full gradients by exploiting the availability of stochastic gradients.

In stochastic optimization, we have access to the stochastic gradient, which is an unbiased estimate of the full gradient [14]. Similar to the case in deterministic optimization, if the objective function is convex and Lipschitz continuous, stochastic gradient (subgradient) descent is the optimal algorithm and the iteration complexity is also $O(\frac{1}{\epsilon^2})$ [14, 15]. When the objective function is $\lambda$-strongly convex, the algorithms proposed in very recent works [9, 10, 21, 26] achieve the optimal $O(\frac{1}{\lambda\epsilon})$ iteration complexity [1]. Since the convergence rate of stochastic optimization is dominated by the randomness in the gradient [6, 11], smoothness usually does not lead to a faster convergence rate for stochastic optimization. A variant of stochastic optimization is the "semi-stochastic" approximation, which interleave stochastic gradient descent and full gradient descent [12]. In the strongly convex case, if the stochastic gradients are taken at a decreasing rate, the convergence rate can be improved to approach $O(\frac{1}{\lambda\sqrt{\epsilon}})$ [13].

From the above discussion, we observe that the iteration complexity in stochastic optimization is polynomial in $\frac{1}{\epsilon}$, making it difficult to find high-precision solutions. However, when the objective function is strongly convex and can be written as a sum of a finite number of functions, i.e.,

$$F(\mathbf{w}) = \frac{1}{n}\sum_{i=1}^{n} f_i(\mathbf{w}), \tag{1}$$

where each $f_i(\cdot)$ is smooth, the iteration complexity of some specific algorithms may exhibit a logarithmic dependence on $\frac{1}{\epsilon}$, i.e., a linear convergence rate. The two very recent works are the stochastic average gradient (SAG) [22], whose iteration complexity is $O(n \log \frac{1}{\epsilon})$, provided $n \geq 8\kappa$, and the stochastic dual coordinate ascent (SDCA) [23], whose iteration complexity is $O((n + \kappa) \log \frac{1}{\epsilon})$.[1] Under approximate conditions, the incremental gradient method [2] and the hybrid method [5] can also minimize the function in (1) with a linear convergence rate. But those algorithms usually treat one pass of all $f_i$'s (or the subset of $f_i$'s) as one iteration, and thus have high computational cost per iteration.

## 3 Epoch Mixed Gradient Descent

### 3.1 Preliminaries

In this paper, we assume there exist two oracles.

1. The first one is a gradient oracle $\mathcal{O}_g$, which for a given input point $\mathbf{w} \in \mathcal{W}$ returns the gradient $\nabla F(\mathbf{w})$, that is,
$$\mathcal{O}_g(\mathbf{w}) = \nabla F(\mathbf{w}).$$

2. The second one is a function oracle $\mathcal{O}_f$, each call of which returns a random function $f(\cdot)$, such that
$$F(\mathbf{w}) = \mathrm{E}_f[f(\mathbf{w})], \ \forall \mathbf{w} \in \mathcal{W},$$
and $f(\cdot)$ is $L$-smooth, that is,
$$\|\nabla f(\mathbf{w}) - \nabla f(\mathbf{w}')\| \leq L\|\mathbf{w} - \mathbf{w}'\|, \ \forall \mathbf{w}, \mathbf{w}' \in \mathcal{W}. \tag{2}$$

Although we do not define a stochastic gradient oracle directly, the function oracle $\mathcal{O}_f$ allows us to evaluate the stochastic gradient of $F(\cdot)$ at any point $\mathbf{w} \in \mathcal{W}$.

Notice that the assumption about the function oracle $\mathcal{O}_f$ implies that the objective function $F(\cdot)$ is also $L$-smooth. Since $\nabla F(\mathbf{w}) = \mathrm{E}_f \nabla f(\mathbf{w})$, by Jensen's inequality, we have

$$\|\nabla F(\mathbf{w}) - \nabla F(\mathbf{w}')\| \leq \mathrm{E}_f \|\nabla f(\mathbf{w}) - \nabla f(\mathbf{w}')\| \stackrel{(2)}{\leq} L\|\mathbf{w} - \mathbf{w}'\|, \ \forall \mathbf{w}, \mathbf{w}' \in \mathcal{W}. \tag{3}$$

Besides, we further assume $F(\cdot)$ is $\lambda$-strongly convex, that is,

$$\|\nabla F(\mathbf{w}) - \nabla F(\mathbf{w}')\| \geq \lambda\|\mathbf{w} - \mathbf{w}'\|, \ \forall \mathbf{w}, \mathbf{w}' \in \mathcal{W}. \tag{4}$$

From (3) and (4), it is obvious that $L \geq \lambda$. The condition number $\kappa$ is defined as the ratio between them. i.e., $\kappa = L/\lambda \geq 1$.

### 3.2 The Algorithm

The detailed steps of the proposed Epoch Mixed Gradient Descent (EMGD) are shown in Algorithm 1, where we use the superscript for the index of epochs, and the subscript for the index of iterations at each epoch. We denote by $\mathcal{B}(\mathbf{x}; r)$ the $\ell_2$ ball of radius $r$ around the point $\mathbf{x}$.

Similar to the epoch gradient descent (EGD) [9], we divided the optimization process into a sequence of epochs (step 3 to step 10). While the number of accesses to the gradient oracle in EGD increases exponentially over the epoches, the number of accesses to the two oracles in EMGD is fixed.

**Algorithm 1** Epoch Mixed Gradient Descent (EMGD)

---

**Input:** step size $\eta$, the initial domain size $\Delta^1$, the number of iterations $T$ per epoch, and the number of epochs $m$

1: Initialize $\bar{\mathbf{w}}^1 = \mathbf{0}$
2: **for** $k = 1, \ldots, m$ **do**
3:     Set $\mathbf{w}_1^k = \bar{\mathbf{w}}^k$
4:     Call the gradient oracle $\mathcal{O}_g$ to obtain $\nabla F(\bar{\mathbf{w}}^k)$
5:     **for** $t = 1, \ldots, T$ **do**
6:         Call the function oracle $\mathcal{O}_f$ to obtain a random function $f_t^k(\cdot)$
7:         Compute the mixed gradient as

$$\tilde{\mathbf{g}}_t^k = \nabla F(\bar{\mathbf{w}}^k) + \nabla f_t^k(\mathbf{w}_t^k) - \nabla f_t^k(\bar{\mathbf{w}}^k)$$

8:         Update the solution by

$$\mathbf{w}_{t+1}^k = \operatorname*{argmin}_{\mathbf{w} \in \mathcal{W} \cap \mathcal{B}(\bar{\mathbf{w}}^k; \Delta^k)} \eta \langle \mathbf{w} - \mathbf{w}_t^k, \tilde{\mathbf{g}}_t^k \rangle + \frac{1}{2}\|\mathbf{w} - \mathbf{w}_t^k\|^2$$

9:     **end for**
10:    Set $\bar{\mathbf{w}}^{k+1} = \frac{1}{T+1}\sum_{t=1}^{T+1} \mathbf{w}_t^k$ and $\Delta^{k+1} = \Delta^k/\sqrt{2}$
11: **end for**
**Return** $\bar{\mathbf{w}}^{m+1}$

---

At the beginning of each epoch, we initialize the solution $\mathbf{w}_1^k$ to be the average solution $\bar{\mathbf{w}}^k$ obtained from the last epoch, and then call the gradient oracle $\mathcal{O}_g$ to obtain $\nabla F(\bar{\mathbf{w}}^k)$. At each iteration $t$ of epoch $k$, we call the function oracle $\mathcal{O}_f$ to obtain a random function $f_t^k(\cdot)$ and define the *mixed gradient* at the current solution $\mathbf{w}_t^k$ as

$$\tilde{\mathbf{g}}_t^k = \nabla F(\bar{\mathbf{w}}^k) + \nabla f_t^k(\mathbf{w}_t^k) - \nabla f_t^k(\bar{\mathbf{w}}^k),$$

which involves both the full gradient and the stochastic gradient. The mixed gradient can be divided into two parts: the deterministic part $\nabla F(\bar{\mathbf{w}}^k)$ and the stochastic part $\nabla f_t^k(\mathbf{w}_t^k) - \nabla f_t^k(\bar{\mathbf{w}}^k)$. Due to the smoothness property of $f_t^k(\cdot)$ and the shrinkage of the domain size, the norm of the stochastic part is well bounded, which is the reason why our algorithm can achieve linear convergence.

Based on the mixed gradient, we update $\mathbf{w}_t^k$ by a gradient mapping over a shrinking domain (i.e., $\mathcal{W} \cap \mathcal{B}(\bar{\mathbf{w}}^k; \Delta^k)$) in step 8. Since the updating is similar to the standard gradient descent except for the domain constraint, we refer to it as mixed gradient descent for short. At the end of the iteration for epoch $k$, we compute the average value of $T + 1$ solutions, instead of $T$ solutions, and update the domain size by reducing a factor of $\sqrt{2}$.

### 3.3 The Convergence Rate

The following theorem shows the convergence rate of the proposed algorithm.

**Theorem 1.** *Assume*

$$\delta \leq e^{-1/2}, \ T \geq \frac{1152 L^2}{\lambda^2} \ln\frac{1}{\delta}, \ and \ \Delta^1 \geq \max\sqrt{\frac{2}{\lambda}(F(\mathbf{0}) - F(\mathbf{w}^*))}. \tag{5}$$

*Set $\eta = 1/[L\sqrt{T}]$. Let $\bar{\mathbf{w}}^{m+1}$ be the solution returned by Algorithm 1 after $m$ epoches that has $m$ accesses to oracle $\mathcal{O}_g$ and $mT$ accesses to oracle $\mathcal{O}_f$. Then, with a probability at least $1 - m\delta$, we have*

$$F(\bar{\mathbf{w}}^{m+1}) - F(\mathbf{w}^*) \leq \frac{\lambda[\Delta^1]^2}{2^{m+1}}, \ and \ \|\bar{\mathbf{w}}^{m+1} - \mathbf{w}^*\|^2 \leq \frac{[\Delta^1]^2}{2^m}.$$

Theorem 1 immediately implies that EMGD is able to achieve an $\epsilon$ optimization error by computing $O(\log\frac{1}{\epsilon})$ full gradients and $O(\kappa^2 \log\frac{1}{\epsilon})$ stochastic gradients.

Table 2: The computational complexity for minimizing $\frac{1}{n}\sum_{i=1}^{n} f_i(\mathbf{w})$

| Nesterov's algorithm [17] | EMGD | SAG ($n \geq 8\kappa$) [22] | SDCA [23] |
|---|---|---|---|
| $O\left(\sqrt{\kappa}n\log\frac{1}{\epsilon}\right)$ | $O\left((n+\kappa^2)\log\frac{1}{\epsilon}\right)$ | $O\left(n\log\frac{1}{\epsilon}\right)$ | $O\left((n+\kappa)\log\frac{1}{\epsilon}\right)$ |

## 3.4 Comparisons

Compared to the optimization algorithms that only rely on full gradients [17], the number of full gradients needed in EMGD is $O(\log\frac{1}{\epsilon})$ instead of $O(\sqrt{\kappa}\log\frac{1}{\epsilon})$. Compared to the optimization algorithms that only rely on stochastic gradients [9,10,21], EMGD is more efficient since it achieves a linear convergence rate.

The proposed EMGD algorithm can also be applied to the special optimization problem considered in [22, 23], where $F(\mathbf{w}) = \frac{1}{n}\sum_{i=1}^{n} f_i(\mathbf{w})$. To make quantitative comparisons, let's assume the full gradient is $n$ times more expensive to compute than the stochastic gradient. Table 2 lists the computational complexities of the algorithms that enjoy linear convergence. As can be seen, the computational complexity of EMGD is lower than Nesterov's algorithm [17] as long as the condition number $\kappa \leq n^{2/3}$, the complexity of SAG [22] is lower than Nesterov's algorithm if $\kappa \leq n/8$, and the complexity of SDCA [23] is lower than Nesterov's algorithm if $\kappa \leq n^2$.[2] The complexity of EMGD is on the same order as SAG and SDCA when $\kappa \leq n^{1/2}$, but higher in other cases. Thus, in terms of computational cost, EMGD may not be the best one, but it has advantages in other aspects.

1. Unlike SAG and SDCA that only work for unconstrained optimization problem, the proposed algorithm works for both constrained and unconstrained optimization problems, provided that the constrained problem in Step 8 can be solved efficiently.

2. Unlike the SAG and SDCA that require an $\Omega(n)$ storage space, the proposed algorithm only requires the storage space of $\Omega(d)$, where $d$ is the dimension of $\mathbf{w}$.

3. The only step in Algorithm 1 that has dependence on $n$ is step 4 for computing the gradient $\nabla F(\bar{\mathbf{w}}^k)$. By utilizing distributed computing, the running time of this step can be reduced to $O(n/k)$, where $k$ is the number of computers, and the convergence rate remains the same. For SAG and SDCA , it is unclear whether they can reduce the running time without affecting the convergence rate.

4. The linear convergence of SAG and SDCA only holds in expectation, whereas the linear convergence of EMGD holds with a high probability, which is much stronger.

## 4 The Analysis

In the proof, we frequently use the following property of strongly convex functions [9].

**Lemma 1.** *Let $f(\mathbf{x})$ be a $\lambda$-strongly convex function over the domain $\mathcal{X}$, and $\mathbf{x}^* = \operatorname{argmin}_{\mathbf{x}\in\mathcal{X}} f(\mathbf{x})$. Then, for any $\mathbf{x} \in \mathcal{X}$, we have*

$$f(\mathbf{x}) - f(\mathbf{x}^*) \geq \frac{\lambda}{2}\|\mathbf{x} - \mathbf{x}^*\|^2. \tag{6}$$

### 4.1 The Main Idea

The Proof of Theorem 1 is based on induction. From the assumption about $\Delta^1$ in (5), we have

$$F(\bar{\mathbf{w}}^1) - F(\mathbf{w}^*) \overset{(5)}{\leq} \frac{\lambda[\Delta^1]^2}{2}, \text{ and } \|\bar{\mathbf{w}}^1 - \mathbf{w}^*\|^2 \overset{(5),(6)}{\leq} [\Delta^1]^2,$$

which means Theorem 1 is true for $m = 0$. Suppose Theorem 1 is true for $m = k$. That is, with a probability at least $1 - k\delta$, we have

$$F(\bar{\mathbf{w}}^{k+1}) - F(\mathbf{w}^*) \leq \frac{\lambda[\Delta^1]^2}{2^{k+1}}, \text{ and } \|\bar{\mathbf{w}}^{k+1} - \mathbf{w}^*\|^2 \leq \frac{[\Delta^1]^2}{2^k}.$$

Our goal is to show that after running the $k+1$-th epoch, with a probability at least $1 - (k+1)\delta$, we have

$$F(\bar{\mathbf{w}}^{k+2}) - F(\mathbf{w}^*) \leq \frac{\lambda[\Delta^1]^2}{2^{k+2}}, \text{ and } \|\bar{\mathbf{w}}^{k+2} - \mathbf{w}^*\|^2 \leq \frac{[\Delta^1]^2}{2^{k+1}}.$$

## 4.2   The Details

For the simplicity of presentation, we drop the index $k$ for epoch. Let $\bar{\mathbf{w}}$ be the solution obtained from the epoch $k$. Given the condition

$$F(\bar{\mathbf{w}}) - F(\mathbf{w}^*) \leq \frac{\lambda}{2}\Delta^2, \text{ and } \|\bar{\mathbf{w}} - \mathbf{w}^*\|^2 \leq \Delta^2, \tag{7}$$

we will show that after running the $T$ iterations in one epoch, the new solution, denoted by $\widehat{\mathbf{w}}$, satisfies

$$F(\widehat{\mathbf{w}}) - F(\mathbf{w}^*) \leq \frac{\lambda}{4}\Delta^2, \text{ and } \|\widehat{\mathbf{w}} - \mathbf{w}^*\|^2 \leq \frac{1}{2}\Delta^2, \tag{8}$$

with a probability at least $1 - \delta$.

Define

$$\mathbf{g} = \nabla F(\bar{\mathbf{w}}), \ \widehat{F}(\mathbf{w}) = F(\mathbf{w}) - \langle\mathbf{w}, \mathbf{g}\rangle, \text{ and } g_t(\mathbf{w}) = f_t(\mathbf{w}) - \langle\mathbf{w}, \nabla f_t(\bar{\mathbf{w}})\rangle. \tag{9}$$

The objective function can be rewritten as

$$F(\mathbf{w}) = \langle\mathbf{w}, \mathbf{g}\rangle + \widehat{F}(\mathbf{w}). \tag{10}$$

And the mixed gradient can be rewritten as

$$\tilde{\mathbf{g}}^k = \mathbf{g} + \nabla g_t(\mathbf{w}_t).$$

Then, the updating rule given in Algorithm 1 becomes

$$\mathbf{w}_{t+1} = \underset{\mathbf{w}\in\mathcal{W}\cap\mathcal{B}(\bar{\mathbf{w}},\Delta)}{\operatorname{argmin}} \eta\langle\mathbf{w} - \mathbf{w}_t, \mathbf{g} + \nabla g_t(\mathbf{w}_t)\rangle + \frac{1}{2}\|\mathbf{w} - \mathbf{w}_t\|^2. \tag{11}$$

Notice that the objective function in (11) is 1-strongly convex. Using the fact that $\mathbf{w}^* \in \mathcal{W} \cap \mathcal{B}(\bar{\mathbf{w}}; \Delta)$ and Lemma 1 (with $\mathbf{x}^* = \mathbf{w}_{t+1}$ and $\mathbf{x} = \mathbf{w}^*$), we have

$$\eta\langle\mathbf{w}_{t+1} - \mathbf{w}_t, \mathbf{g} + \nabla g_t(\mathbf{w}_t)\rangle + \frac{1}{2}\|\mathbf{w}_{t+1} - \mathbf{w}_t\|^2$$

$$\leq \eta\langle\mathbf{w}^* - \mathbf{w}_t, \mathbf{g} + \nabla g_t(\mathbf{w}_t)\rangle + \frac{1}{2}\|\mathbf{w}^* - \mathbf{w}_t\|^2 - \frac{1}{2}\|\mathbf{w}^* - \mathbf{w}_{t+1}\|^2. \tag{12}$$

For each iteration $t$ in the current epoch, we have

$$F(\mathbf{w}_t) - F(\mathbf{w}^*)$$

$$\overset{(4)}{\leq} \langle\nabla F(\mathbf{w}_t), \mathbf{w}_t - \mathbf{w}^*\rangle - \frac{\lambda}{2}\|\mathbf{w}_t - \mathbf{w}^*\|^2 \tag{13}$$

$$\overset{(10)}{=} \langle\mathbf{g} + \nabla g_t(\mathbf{w}_t), \mathbf{w}_t - \mathbf{w}^*\rangle + \left\langle\nabla\widehat{F}(\mathbf{w}_t) - \nabla g_t(\mathbf{w}_t), \mathbf{w}_t - \mathbf{w}^*\right\rangle - \frac{\lambda}{2}\|\mathbf{w}_t - \mathbf{w}^*\|^2,$$

and

$$\langle\mathbf{g} + \nabla g_t(\mathbf{w}_t), \mathbf{w}_t - \mathbf{w}^*\rangle$$

$$\overset{(12)}{\leq} \langle\mathbf{g} + \nabla g_t(\mathbf{w}_t), \mathbf{w}_t - \mathbf{w}_{t+1}\rangle + \frac{\|\mathbf{w}_t - \mathbf{w}^*\|^2}{2\eta} - \frac{\|\mathbf{w}_{t+1} - \mathbf{w}^*\|^2}{2\eta} - \frac{\|\mathbf{w}_t - \mathbf{w}_{t+1}\|^2}{2\eta}$$

$$\leq \langle\mathbf{g}, \mathbf{w}_t - \mathbf{w}_{t+1}\rangle + \frac{\|\mathbf{w}_t - \mathbf{w}^*\|^2}{2\eta} - \frac{\|\mathbf{w}_{t+1} - \mathbf{w}^*\|^2}{2\eta} \tag{14}$$

$$+ \max_{\mathbf{w}}\left(\langle\nabla g_t(\mathbf{w}_t), \mathbf{w}_t - \mathbf{w}\rangle - \frac{\|\mathbf{w}_t - \mathbf{w}\|^2}{2\eta}\right)$$

$$= \langle\mathbf{g}, \mathbf{w}_t - \mathbf{w}_{t+1}\rangle + \frac{\|\mathbf{w}_t - \mathbf{w}^*\|^2}{2\eta} - \frac{\|\mathbf{w}_{t+1} - \mathbf{w}^*\|^2}{2\eta} + \frac{\eta}{2}\|\nabla g_t(\mathbf{w}_t)\|^2.$$

Combining (13) and (14), we have

$$F(\mathbf{w}_t) - F(\mathbf{w}^*)$$

$$\leq \frac{\|\mathbf{w}_t - \mathbf{w}^*\|^2}{2\eta} - \frac{\|\mathbf{w}_{t+1} - \mathbf{w}^*\|^2}{2\eta} - \frac{\lambda}{2}\|\mathbf{w}_t - \mathbf{w}^*\|^2$$

$$+ \langle \mathbf{g}, \mathbf{w}_t - \mathbf{w}_{t+1}\rangle + \frac{\eta}{2}\|\nabla g_t(\mathbf{w}_t)\|^2 + \left\langle \nabla \widehat{F}(\mathbf{w}_t) - \nabla g_t(\mathbf{w}_t), \mathbf{w}_t - \mathbf{w}^*\right\rangle.$$

By adding the inequalities of all iterations, we have

$$\sum_{t=1}^{T}(F(\mathbf{w}_t) - F(\mathbf{w}^*))$$

$$\leq \frac{\|\bar{\mathbf{w}} - \mathbf{w}^*\|^2}{2\eta} - \frac{\|\mathbf{w}_{T+1} - \mathbf{w}^*\|^2}{2\eta} - \frac{\lambda}{2}\sum_{t=1}^{T}\|\mathbf{w}_t - \mathbf{w}^*\|^2 + \langle \mathbf{g}, \bar{\mathbf{w}} - \mathbf{w}_{T+1}\rangle \quad (15)$$

$$+ \underbrace{\frac{\eta}{2}\sum_{t=1}^{T}\|\nabla g_t(\mathbf{w}_t)\|^2}_{\triangleq A_T} + \underbrace{\sum_{t=1}^{T}\langle \nabla \widehat{F}(\mathbf{w}_t) - \nabla g_t(\mathbf{w}_t), \mathbf{w}_t - \mathbf{w}^*\rangle}_{\triangleq B_T}.$$

Since $F(\cdot)$ is $L$-smooth, we have

$$F(\mathbf{w}_{T+1}) - F(\bar{\mathbf{w}}) \leq \langle \nabla F(\bar{\mathbf{w}}), \mathbf{w}_{T+1} - \bar{\mathbf{w}}\rangle + \frac{L}{2}\|\bar{\mathbf{w}} - \mathbf{w}_{T+1}\|^2,$$

which implies

$$\langle \mathbf{g}, \bar{\mathbf{w}} - \mathbf{w}_{T+1}\rangle \leq F(\bar{\mathbf{w}}) - F(\mathbf{w}_{T+1}) + \frac{L}{2}\Delta^2$$
$$\overset{(7)}{\leq} F(\mathbf{w}^*) - F(\mathbf{w}_{T+1}) + \frac{\lambda}{2}\Delta^2 + \frac{L}{2}\Delta^2 \leq F(\mathbf{w}^*) - F(\mathbf{w}_{T+1}) + L\Delta^2. \quad (16)$$

From (15) and (16), we have

$$\sum_{t=1}^{T+1}(F(\mathbf{w}_t) - F(\mathbf{w}^*)) \leq \Delta^2\left(\frac{1}{2\eta} + L\right) + \frac{\eta}{2}A_T + B_T. \quad (17)$$

Next, we consider how to bound $A_T$ and $B_T$. The upper bound of $A_T$ is given by

$$A_T = \sum_{t=1}^{T}\|\nabla g_t(\mathbf{w}_t)\|^2 = \sum_{t=1}^{T}\|\nabla f_t(\mathbf{w}_t) - \nabla f_t(\bar{\mathbf{w}})\|^2 \overset{(2)}{\leq} L^2\sum_{t=1}^{T}\|\mathbf{w}_t - \bar{\mathbf{w}}\|^2 \leq TL^2\Delta^2. \quad (18)$$

To bound $B_T$, we need the Hoeffding-Azuma inequality stated below [4].

**Lemma 2.** *Let $V_1, V_2, \ldots$ be a martingale difference sequence with respect to some sequence $X_1, X_2, \ldots$ such that $V_i \in [A_i, A_i + c_i]$ for some random variable $A_i$, measurable with respect to $X_1, \ldots, X_{i-1}$ and a positive constant $c_i$. If $S_n = \sum_{i=1}^{n}V_i$, then for any $t > 0$,*

$$\Pr[S_n > t] \leq \exp\left(-\frac{2t^2}{\sum_{i=1}^{n}c_i^2}\right).$$

Define

$$V_t = \langle \nabla \widehat{F}(\mathbf{w}_t) - \nabla g_t(\mathbf{w}_t), \mathbf{w}_t - \mathbf{w}^*\rangle, \ t = 1, \ldots, T.$$

Recall the definition of $\widehat{F}(\cdot)$ and $g_t(\cdot)$ in (9). Based on our assumption about the function oracle $\mathcal{O}_f$, it is straightforward to check that $V_1, \ldots$ is a martingale difference with respect to $g_1, \ldots$. The value of $V_t$ can be bounded by

$$
\begin{aligned}
|V_t| &\leq \left\|\nabla \widehat{F}(\mathbf{w}_t) - \nabla g_t(\mathbf{w}_t)\right\|\|\mathbf{w}_t - \mathbf{w}^*\| \\
&\leq 2\Delta\left(\|\nabla F(\mathbf{w}_t) - \nabla F(\bar{\mathbf{w}})\| + \|\nabla f_t(\mathbf{w}_t) - \nabla f_t(\bar{\mathbf{w}})\|\right) \\
&\overset{(2),(3)}{\leq} 4L\Delta\|\mathbf{w}_t - \bar{\mathbf{w}}\| \leq 4L\Delta^2.
\end{aligned}
$$

Following Lemma 2, with a probability at least $1 - \delta$, we have

$$B_T \le 4L\Delta^2 \sqrt{2T \ln \frac{1}{\delta}}. \tag{19}$$

By adding the inequalities in (17), (18) and (19) together, with a probability at least $1 - \delta$, we have

$$\sum_{t=1}^{T+1} \left( F(\mathbf{w}_t) - F(\mathbf{w}^*) \right) \le \Delta^2 \left( \frac{1}{2\eta} + L + \frac{\eta T L^2}{2} + 4L\sqrt{2T \ln \frac{1}{\delta}} \right).$$

By choosing $\eta = 1/[L\sqrt{T}]$, we have

$$\sum_{t=1}^{T+1} \left( F(\mathbf{w}_t) - F(\mathbf{w}^*) \right) \le L\Delta^2 \left( \sqrt{T} + 1 + 4\sqrt{2T \ln \frac{1}{\delta}} \right) \overset{(5)}{\le} 6L\Delta^2 \sqrt{2T \ln \frac{1}{\delta}}, \tag{20}$$

where in the second inequality we use the condition $\delta \le e^{-1/2}$ in (5). By Jensen's inequality, we have

$$F(\widehat{\mathbf{w}}) - F(\mathbf{w}^*) \le \frac{1}{T+1} \sum_{t=1}^{T+1} \left( F(\mathbf{w}_t) - F(\mathbf{w}^*) \right) \overset{(20)}{\le} \Delta^2 \frac{6L\sqrt{2\ln 1/\delta}}{\sqrt{T+1}},$$

and therefore

$$\|\widehat{\mathbf{w}} - \mathbf{w}^*\|^2 \overset{(6)}{\le} \frac{2}{\lambda} F(\widehat{\mathbf{w}}) - F(\mathbf{w}^*) \le \Delta^2 \frac{12L\sqrt{2\ln 1/\delta}}{\lambda\sqrt{T+1}}.$$

Thus, when

$$T \ge \frac{1152L^2}{\lambda^2} \ln \frac{1}{\delta},$$

with a probability at least $1 - \delta$, we have

$$F(\widehat{\mathbf{w}}) - F(\mathbf{w}^*) \le \frac{\lambda}{4}\Delta^2, \text{ and } \|\widehat{\mathbf{w}} - \mathbf{w}^*\|^2 \le \frac{1}{2}\Delta^2.$$

## 5  Conclusion and Future Work

In this paper, we consider how to reduce the number of full gradients needed for smooth and strongly convex optimization problems. Under the assumption that both the gradient and the stochastic gradient are available, a novel algorithm named Epoch Mixed Gradient Descent (EMGD) is proposed. Theoretical analysis shows that with the help of stochastic gradients, we are able to reduce the number of gradients needed from $O(\sqrt{\kappa} \log \frac{1}{\epsilon})$ to $O(\log \frac{1}{\epsilon})$. In the case that the objective function is in the form of (1), i.e., a sum of $n$ smooth functions, EMGD has lower computational cost than the full gradient method [17], if the condition number $\kappa \le n^{2/3}$.

In practice, a drawback of EMGD is that it requires the condition number $\kappa$ is known beforehand. We will interstage how to find a good estimation of $\kappa$ in future. When the objective function is a sum of some special functions, such as the square loss (i.e., $(y_i - \mathbf{x}_i^\top \mathbf{w})^2$), we can estimate the condition number by sampling. In particular, the Hessian matrix estimated from a subset of functions, combined with the concentration inequalities for matrix [7], can be used to bound the eigenvalues of the true Hessian matrix and consequentially $\kappa$. Furthermore, if there exists a strongly convex regularizer in the objective function, which happens in many machine learning problems [8], the knowledge of the regularizer itself allows us to find an upper bound of $\kappa$.

### Acknowledgments

This work is partially supported by ONR Award N000141210431 and NSF (IIS-1251031).

## Footnotes

[1] In order to apply SDCA, we need to assume each function $f_i$ is $\lambda$-strongly convex, so that we can rewrite $f_i(\mathbf{w})$ as $g_i(\mathbf{w}) + \frac{\lambda}{2}\|\mathbf{w}\|^2$, where $g_i(\mathbf{w}) = f_i(\mathbf{w}) - \frac{\lambda}{2}\|\mathbf{w}\|^2$ is convex.

[2]In machine learning, we usually face a regularized optimization problem $\min_{\mathbf{w}\in\mathcal{W}} \frac{1}{n}\sum_{i=1}^{n} \ell(y_i; \mathbf{x}_i^\top\mathbf{w}) + \frac{\tau}{2}\|\mathbf{w}\|^2$, where $\ell(\cdot;\cdot)$ is some loss function. When the norm of the data is bounded, the smoothness parameter $L$ can be treated as a constant. The strong convexity parameter $\lambda$ is lower bounded by $\tau$. Thus, as long as $\tau > \Omega(n^{-2/3})$, which is a reasonable scenario [25], we have $\kappa < O(n^{2/3})$, indicating our proposed EMGD can be applied.

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
