[Reviews · NeurIPS 2013]

Submitted by Assigned_Reviewer_5

quality: 7 (out of 10)
clarity: 8
originality: 8
significance: 7

SUMMARY: The authors consider the problem of optimizing a smooth and strongly convex function over a convex constraint set such that the gradient mapping update can be computed efficiently. The optimal first-order algorithm of Nesterov has linear convergence for such problem but the constant depends on the square root of the condition number k. The authors consider the situation where one has access to the expensive full gradient of the objective as well as a cheap stochastic gradient oracle. They propose a hybrid algorithm which only requires O(log 1/eps) calls to the full gradient oracle (independent of the condition number) and O(k^2 log(1/eps)) calls to the cheaper stochastic gradient oracle -- as long as the condition number is not too big, this could be faster in theory. The main idea behind their algorithm(called Epoch Mixed Gradient Descent - EMGD) is to replace a full gradient step (called an epoch) with a fixed number O(k^2) of mixed gradient steps which use a combination of the full gradient (computed once for the epoch) and stochastic gradients (which vary within an epoch). By taking the average of the O(k^2) iterates within an epoch, they can show a constant decrease of the suboptimality *independent* of the condition number, which is why the number of required full gradient step computations (the number of epochs) is independent from the condition number. They provide a simple and complete self-contained proof of their convergence rate, but no experiment.

I enjoyed reading this paper and I think that the idea of mixing a few full gradient computations with a large number of cheap stochastic gradient steps is novel and interesting. This work situates itself in a string of recent papers which attempt to use the cheaper stochastic oracle while maintaining a linear convergence rate. A recent theoretical AND practical breakthrough was made with the SAG algorithm [16] which works for a smooth strongly convex objective which is the sum of n simple functions (such as in regularized empirical loss minimization - where n is the number of training examples). In this case, a reasonable assumption is that the full gradient oracle is n times more expensive to compute than the stochastic gradient one. Then SAG is faster in theory to Nesterov' algorithm as long as the condition number k < = n/8. In contrast, EMGD in this case is faster to Nesterov' algorithm in theory as long as the condition number k < = n^(2/3), and EMGD is slower than SAG in all regimes (it has the same big O when k < = n^1/2). To get a sense of these speed-ups, if k = n^1/2, then both SAG & EMGD are O(n log(1/eps)) whereas Nesterov' algorithm O(n^(5/4) log(1/eps)). The two main advantages that I see for EMGD over SAG are that 1) as mentioned by the authors, EMGD works for constrained optimization [supposing that the gradient mapping update can be computed efficiently] whereas SAG is only defined so far for unconstrained problems; and 2) the convergence proof for EMGD is much simpler than the one for SAG and so could yield more insights as well as make the modifications to EMGD more amenable to provable guarantees [their result is also stronger as it holds with high probability vs. in expectation for SAG]. The authors also mention a possible memory / parallelization advantage, though this is less clear as SAG can also be parallelized using mini-batches (which also reduces the memory requirement by the size of the mini-batches).

EVALUATION SUMMARY:
Pros:
- Gives a novel and interesting algorithmic idea with a clean, simple and solid theory.
- Like SAG, get a linear convergence rate for regularized empirical loss minimization where the condition number k is not multiplying the number of training examples n in the constant; but their algorithm is more general (works for constrained optimization) and the proof is much simpler.
- The paper is clearly written and the proof is self-contained.

Cons:
- There is no experiment which could show that this algorithm could actually make a difference in practice (and doesn't provide any concrete example to illustrate why its suggested theoretical advantages could be relevant in practice).
- There is no discussion of the limitations / drawbacks of the algorithm (especially, in comparison to the existing algorithms -- section 3.4 should be improved! I make several suggestions in this review).
- The proof is lacking some high-level comments which could justify the essential insights used to its construction.

QUALITY: The paper is technically sound. Some experiments would have been appreciated, though I think that the theoretical contribution could stand on its own. The authors should definitively extend section 3.4 with the limitations and drawbacks of their algorithm though. They should also add a more concrete discussion of the sum of n functions example which highlights nicely the differences with SAG and Nesterov (as I mentioned above -- e.g. EMGD is worse than Nesterov for (roughly) k > n^(2/3); SAG is same as EMGD for 1 < = k < = n^(1/2); better than EMGD (and Nesterov) for n^(1/2) < k < = n/8; and worse than Nesterov for k > n/8). A major drawback of the EMGD for the practitioner is that the number of steps within an epoch needs to be fixed in advance (with the knowledge of k) -- in contrast, both SCDA [17] and SAG don't have to fix the number of steps in advance and so can benefit for having a faster practical convergence than the bound would predict (or benefit from better local condition number e.g.). Moreover, SCDA has automatic step-size selection; whereas SAG has an adaptive step-size heuristic which seems to work quite well in practice. The authors should add this discussion in the paper.
** [ADDENDUM after discussion with other reviewers: a) As another reviewer mentioned, often in practice in ML the regularization parameter is C/n and so the condition number is ~= n/C', which is a regime in which SAG still does better than Nesterov, but in which EMGD *doesn't*. This should be pointed out (and perhaps another practical setting where k < n^(2/3) should mentioned). b) The authors should cite [Hybrid Deterministic-Stochastic Methods for Data Fitting. M. Friedlander, M. Schmidt. SISC, 2012] which also presents a hybrid deterministic-stochastic algorithm with a *linear* rate of convergence. This latter algorithm still has the condition number appearing in the rate though, so the ultimate rate is not faster than standard gradient descent (just the beginning is faster because of using cheaper steps) -- so the current submission can still improve theoretically over this rate when k is not too big with respect to n.] **


CLARITY: The paper is fairly clear. I have appreciated the summary from Table 1 which I haven't seen in the literature in such a clear manner. The proof can be followed tightly, but would be more useful to the reader if the authors could add a few high-level comments motivating some of the defined quantities which are fitting together a bit too magically in its current form.

ORIGINALITY: This is a novel combination of known techniques.

SIGNIFICANCE: The practical relevance of the algorithm is not demonstrated yet. But the theoretical contribution could have impact. In the context of the difficult proof of convergence of SAG (and the simple proof of SDCA, but which only applies to a restricted setup), the simple proof of EMGD is a major contribution.

== More detailed suggestions ==

- line 050: I suggest to say "are summarized in Table 1 and detailed in the related work section". One can wonder at first what are the citations for these rates.

- line 118-119: I would specify here that SCDA is only defined for a specific form of f_i, unlike SAG which handles any smooth convex f_i.

- line 132: I would explicitly state that this condition is "for all w" -- it was a bit ambiguous whether the condition was only for a fixed w (the 'given input point w' mentioned in 1), which of course would not be sufficient to have the gradients match.

- line 139: I would mention explicitly that we also have \grad F(w) = E[\grad f(w) ].

- line 224: The claim that SAG (or SCDA) cannot take advantage of distributed computing is false: they both can use mini-batches (this can also reduce the memory requirement).

- line 288: I would write "(7) on (10) with x=w* (feasible by (5)) and x*=w_{t+1}" to be more explicit -- it is not that obvious how to obtain the line otherwise...

- line 304 (and all other places): Add parenthesis around (F(w_t)-F(w*)) to be explicit that F(w*) is also summed T times...

- line 366: Add that the 2nd inequality is only valid for all T > = 1 if delta < = exp(-1/2) [this also explains a condition stated in (4) which appeared nowhere explicitly].

Some typos:
-042: 'an convex' - > 'a convex'
-056: *strong* convexity, *condition* number
-088 (and other places): please correct the notation of your domain to b consistent -- you use D in 039 and the first few pages; here, you use \Omega -- choose one and use it everywhere in the paper!
- 307: use norm symbol instead of absolute value
-366: for consistency, replace log with ln in the middle equation


=== Update after rebuttal ===

I am happy with the response by the authors, but note that I will carefully check that their updated version is implementing the changes that we have suggested!
Summary: The authors give a novel and interesting algorithmic idea for smooth convex optimization with a clean, simple and solid theory. The paper is lacking an experimental section to illustrate the practical relevance of the algorithm, but I think that the theoretical contribution can stand on its own, assuming that the authors add a more complete discussion of the properties of the algorithm as I have described in this review.

Submitted by Assigned_Reviewer_6

The paper presents a "hybrid" deterministic-stochastic method for first-order optimization. The algorithm mixes calls to a deterministic oracle (querying full gradients) and stochastic oracle (querying stochastic gradients). The authors show that the approach allows to drop the dependency in the condition number in the rate of convergence.


The authors present an interesting and striking theoretical result. Basically, assuming the condition number is known (i.e. both strong convexity and strong smoothness constants of the objective are known), assuming first-order hybrid oracle (both stochastic and deterministic gradient can be queried) is available, then with a suitably chosen mix of deterministic gradient and stochastic gradient steps, one can achieve with high-probability a O(log(1/epsilon)) convergence rate for batch optimization.

However, there are several concerns with the current state of the paper. First, the theoretical analysis only covers the case where the condition number is perfectly known beforehand, and does not cover the behavior of the algorithm when this hyper-parameter of the algorithm is misspecified. Basically, the current resuIt says that the dependence on the condition number (kappa) can be removed from the convergence rate of a first-order optimization algorithm when this condition number is known before hand to the algorithm. Maybe, the theoretical analysis in these other cases (when kappa is unknown) is challenging. If so, then this analysis could have been conducted through experiments. But the paper has no experiments section, which is the second major concern. Since the main contribution of the paper is a new algorithm, then I guess it would make sense to at least perform some experiments to assess the theoretical results presented in the paper, and study their relevance wrt the actual behavior of the algorithm on empirical data.



Detailed comments

The proposed algorithm (EGD) relies on the update rule defined by Eq. 8 (page 4) using the so-called "mixed gradient" defined in Eq. 7. Therefore, EGD requires $\eta$ as a hyper-parameter to be set (or estimated). Setting $\eta$ boils down to knowing the condition number $\kappa$ ("conditional number" in the paper), that is both the "strong convexity modulus" $\lambda$ and the "strong convexity modulus" $L$. As far as I understood the paper, the authors do not provide any guideline or theoretical argument allowing to set $\kappa$ *beforehand*.

So, there are two possibilities. Either this parameter has to be estimated, and the corresponding estimation procedure is missing (just a heuristic procedure would be fine, as long as it is supported by numerical experiments) . Or, this parameter is assumed known, because the point of the paper is mainly theoretical. But then the theoretical analysis/experimental section should cover the cases where this hyper-parameter is misspecified. This implies studying the convergence rate in cases where the hyper-parameter is set too large or set too small.

Although popular in theoretical analysis, and realistic in many situations, the smooth and strongly convex case can be too restrictive, and other settings (non-strongly convex) are also interesting. In particular, the non-strongly convex case is important as well, as it also arises in several situations. See Bach & Moulines, 2011, for a theoretical analysis of the different behaviors depending on the cases (convex vs strongly convex).

There are other concerns. Blending deterministic gradient steps and stochastic gradient steps in a first-order optimization algorithm is not a new idea. Actually, the algorithm presented in the paper is not written this way, that is as an alternation of deterministic gradient steps and stochastic gradient steps, with different of frequencies for each type of steps. It is written as one "mixed" update per iteration (within an epoch), and then a gradient-like update step. It would be interesting to discuss how the proposed algorithm relates and compares with a similar-in-spirit algorithm where one would interleave deterministic gradient steps and stochastic gradient steps (at least in the "unconstrained" case).

The authors do not review a related line of work, namely so-called hybrid deterministic-stochastic optimization algorithms; see [Hybrid Deterministic-Stochastic Methods for Data Fitting. M. Friedlander, M. Schmidt. SISC, 2012]. Discussing and comparing the convergence rates would be valuable here. See also the above ref. for a review of older works on that topic.

Finally, a thorough experimental study would be a valuable addition to the paper, including a detailed comparison with regular SGD, averaged SGD, and recent proposals for stochastic first-order optimization (SAG, etc.).

A more minor concern, the optimization setting considered in the paper is not clearly stated. The purpose of the paper is to get the best of both worlds (deterministic optimization and stochastic optimization), namely exponential rate of convergence from the deterministic world and dependence on the condition number from the stochastic world. The authors do not specify clearly what they intend to solve: the deterministic optimization problem [Min_w F(w)=1/n \sum_{i=1}^n F_i(w)] with a "stochastic" (or more precisely, "randomized") algorithm, or the stochastic approximation problem [Min_w E(F(w))]. Note in passing that both SAG and SDCA are randomized algorithms for solving the *deterministic* optimization problem [Min_w F(w)=1/n \sum_{i=1}^n F_i(w)]. The theoretical setup stated in Section 3.1 is misleading from this respect, and none of the claims made later in the paper clarifies which setting is considered. This could easily be fixed.
Summary: The authors present an interesting and striking theoretical result. Basically, assuming the condition number is known (i.e. both strong convexity and strong smoothness constants of the objective are known), assuming first-order hybrid oracle (both stochastic and deterministic gradient can be queried) is available, then with a suitably chosen mix of deterministic gradient and stochastic gradient steps, one can achieve with high-probability a O(log(1/epsilon)) convergence rate for batch optimization.

However, there are several concerns with the current state of the paper. First, the theoretical analysis only covers the case where the condition number is perfectly known beforehand. The behavior of the algorithm when this hyper-parameter of the algorithm is misspecified is not discussed. Basically, the current resuIt says that the dependence on the condition number (kappa) can be removed from the convergence rate of a first-order optimization algorithm when this condition number is known before hand to the algorithm. Maybe, the theoretical analysis in these other cases (when kappa is unknown) is challenging. If so, then this analysis could have been conducted through experiments. But the paper has no experiments section, which is the second major concern. Since the main contribution of the paper is a new algorithm, then I guess it would make sense to at least perform some experiments to assess the theoretical results presented in the paper, and study their relevance wrt the actual behavior of the algorithm on empirical data.

Submitted by Assigned_Reviewer_7

The paper presents a new hybrid strategy for stochastic gradient descent, that employs both stochastic and batch gradients. The algorithm is guaranteed to converge to an epsilon accurate solution using O(log(1/eps)) full gradients, and O(k^2 log(1/eps)) stochastic gradients.

The convergence proof appears correct and novel to me, a part for some minor mistakes detailed below.

My main concern is about the relevance of the proposed algorithm in the machine learning setting, that is the focus of the conference.
In fact, in usual ML algorithms the strong convexity is given by the regularizer. Hence, the value of mu is of the order of the number of samples N, that is something like mu = C N, where C does not depend on N. With this assumption, the proposed method is faster than batch gradient only if the number of samples is bounded by O(C^3/L^3), that does not seem to me an interesting regime. Moreover the convergence rate for the proposed algorithm holds only in high probability, while the ones for batch gradient descent is deterministic.
This point is very important and it must be carefully discussed, to actually show that the algorithms has a real advantage over batch gradient descent, and to prove the relevance of the paper for the ML community.

Minor comments:
- equation (6) should be ||w^*-\hat{w}||^2
- please specify in 288 on which function you use (7)
- the equality in 286 should be removed: it adds nothing to the comprehension, rather it decreases it
- in (13) the absolute values should be norms
- Please explain somewhere the fact that L \geq lambda, even if it is obvious, it is better to state it more clearly
- In (4) x^* should be w^* and f should be F, and the first term in the max is always bigger than the second one, by Lemma 1
- Please precisely define the condition number as a function of lambda and L
Summary: Novel hybrid stochastic/batch gradient descent. Not clear if the algorithm has any advantage over standard batch gradient descent in practical ML optimization problems.
Author Feedback

Author rebuttal: Thanks for the comments!

-----For Assigned_Reviewer_5
We will revise our paper based on the detailed suggestions. In particular, we will add more discussions about the limitation of our work, the estimation of the condition number, and the related hybrid deterministic-stochastic methods.

For the two questions in the addendum, please refer to the responses to the other reviewers.

-----For Assigned_Reviewer_6
Q: The authors do not provide any guideline or theoretical argument allowing to set $\kappa$ *beforehand*.
A: We will add more discussions on this issue. When there is a $\mu$-strongly convex regularizer, the condition number can be upper bounded by $L/\mu$, where L is the Lipschitz constant of the gradient. When the strongly convexity arises from special losses, such as the square loss, we can estimate the condition number by sampling. In particular, the empirical Hessian matrix estimated from the sampled training examples, combined with the concentration inequalities for matrix, can be used to estimate the lower and upper bounds of the eigenvalues of true Hessian matrix and consequentially the condition number.

Q: A back-of-the-envelope calculation seems to indicate that one gets an algorithm with similar behavior if one interleaves deterministic gradient steps and stochastic gradient steps (at least in the "unconstrained" case).
A: If we interleave deterministic gradient steps and stochastic gradient steps, the number of full gradient and stochastic gradients will be on the sample order. Based on the lower’s bound of iteration complexity provided by Nesterov, the number of full gradients will depend on the condition number. In contrast, the number of full gradients used by our algorithm is *INDEPENDENT* from the condition number, and the number of stochastic gradients is *INDEPENDENT* from the problem size $n$.

Q: The authors do not review a very related recent line of work, namely so-called hybrid deterministic-stochastic optimization algorithms. … Hybrid algorithms were actually the focus of an important old literature, under the name of "semi-stochastic algorithms" or "hybrid algorithms".

A: We appreciate related work pointed out by the reviewer and will include them in the revised draft. We emphasize that the focus of this work is to optimize a *deterministic* objective function which is both smooth and strongly convex. Our goal is to achieve a linear convergence but with the number of calls to the full gradient oracle independent from the condition number, which is different from all the previous works. The related studies, pointed out by the reviewer, either work under very strong assumption about the stochastic gradient oracle (e.g., “Hybrid Deterministic-Stochastic Methods for Data Fitting. M. Friedlander, M. Schmidt. SISC, 2012”) or do not yield linear convergence (e.g. “Rates of convergence of semi-stochastic approximation procedures for solving stochastic optimization problems”). In addition, as pointed out by the reviewer, none of these studies is able to make the number of calls to the full gradient oracle independent from the condition number.

Q: The optimization setting considered in the paper is not clearly stated.
A: We aim to optimize a *deterministic* objective function under the assumption that both full gradients and stochastic gradients are available. Our goal is to make the number of full gradients independent from the condition number by making use of stochastic gradients.

-----For Assigned_Reviewer_7
Q: In fact, in usual ML algorithms the strong convexity is given by the regularizer. Hence, the value of mu is of the order of the number of samples N, that is something like mu = C N
A: Consider the optimization problem $1/N \sum_{i=1}^N \ell(x_i,y_i;w) + \mu |w|$ frequently faced in machine learning. The condition number may not be small because of the following two reasons:
1) According to the results in learning theory (see Page 1166 of “SVM Soft Margin Classifiers: Linear Programming versus Quadratic Programming, Neural Computation, 2005”), the best order of the $\mu$ ranges from $N^{-1}$ to $N^{-1/2}$. Thus, the condition number can be as small as $N^{1/2}$. As pointed out by the first reviewer, our algorithm is faster than the full gradient methods as long as the condition number is small than $N^{2/3}$.
2) Besides the regularizer, the loss function may also contribute to the strongly convexity. For instance, when the loss function is the square loss or the logit loss, $1/N \sum_{i=1}^N \ell(x_i,y_i;w)$ can be strongly convex when $N$ is significantly larger than the dimensionality.